

# TAGGS: Grouping Tweets to Improve Global Geotagging for Disaster Response

Jens de Bruijn[1], Hans de Moel[1], Brenden Jongman[1,2], Jurjen Wagemaker[3], Jeroen C.J.H. Aerts[1]

[1]Institute for Environmental Studies, VU University, Amsterdam, 1081HV, The Netherlands
5 [2]Global Facility for Disaster Reduction and Recovery, World Bank Group, Washington D.C., 20433, USA
[3]FloodTags, The Hague, 2511 BE, The Netherlands

*Correspondence to*: Jens de Bruijn (j.a.debruijn@outlook.com)

**Abstract.** The availability of timely and accurate information about ongoing events is important for relief organizations seeking to effectively respond to disasters. Recently, social media platforms, and in particular Twitter, have gained traction as 10 a novel source of information on disaster events. Unfortunately, geographical information is rarely attached to tweets, which hinders the use of Twitter for geographical applications. As a solution, analyses of a tweet's text, combined with an evaluation of its metadata, can help to increase the number of geo-located tweets. This paper describes a new algorithm (TAGGS), that georeferences tweets by using the spatial information of groups of tweets mentioning the same location. This technique results in a roughly twofold increase in the number of geo-located tweets as compared to existing methods. We applied this approach 15 to 35.1 million flood-related tweets in 12 languages, collected over 2.5 years. In the dataset, we found 11.6 million tweets mentioning one or more flood locations, which can be towns (6.9 million), provinces (3.3 million), or countries (2.2 million). Validation demonstrated that TAGGS correctly located about 65-75% of the tweets. As a future application, TAGGS could form the basis for a global event detection and monitoring system.

## 1 Introduction

20 Each year, natural disasters affect roughly one million people, causing thousands of deaths, and tens of billions of US dollars in damages (Guha-Sapir et al., 2012). The availability of timely and accurate information about the impacts of an ongoing event can assist relief organizations in enhancing their disaster response activities, and thus mitigate the consequences of disasters (de Perez et al., 2014; Messner & Meyer, 2006; Penning-rowsell et al., 2005). Information about an ongoing event is, however, often difficult to obtain. Such data is generally collected using measurement instruments such as remote sensors 25 (e.g., Sun et al. 2000), well as from both local relief and response professionals and analyses of media reports (Jongman et al., 2015). Recently, social media, and in particular Twitter, has gained traction as a novel source of information on disaster events. The Twitter posts ("tweets") sent out by millions of users around the globe hold great potential in disaster management (Carley et al., 2016; Jongman et al., 2015; Sakaki et al., 2010). When correctly analyzed, they can improve the detection of disasters (Ghahremanlou et al., 2014) and provide valuable information about the societal impacts of ongoing disaster events (Fohringer 30 et al., 2015; Gao et al., 2011; Jongman et al., 2015). In computer science, social media has been studied extensively.



Researchers have also developed several applications for applied geographic research. Examples of such applications include detection systems for flood events and (Jongman et al., 2015) earthquake disasters (Crooks et al., 2013; Sakaki et al., 2010).

One of the key issues in using Twitter information to assess the impacts of natural disasters entails accurately localizing
individual tweets. Twitter allows users to automatically attach their current GPS location to a tweet, specifying their position at the moment a tweet is posted (Sakaki et al., 2010). However, because this feature is turned off by default, only 0.9% of the tweets have coordinate information attached (Lee et al., 2013). Another method of extracting geographical information from a tweet is "geotagging," an approach that employs text and/or tweet metadata to detect a user's residence, the location from which the tweet was sent, or the location to which a tweet refers (Schulz et al., 2013). For detection and monitoring purposes,
data on the location referenced by a tweet is required. A geotagging algorithm extracts and locates the mentioned geographical locations ("toponyms") from a text. Research has demonstrated that geotagging algorithms can dramatically increase the number of geo-located posts (e.g., Schulz et al. 2013; Paradesi 2011; Karimzadeh et al. 2013).

Geotagging has been discussed in numerous studies (Amitay et al., 2004; Ghahremanlou et al., 2014; Lieberman et al., 2010).
This literature domain has identified two distinct steps that comprise geotagging: *toponym recognition* and *toponym resolution* (Leidner, 2007).

The first step is *toponym recognition,* which entails identifying geographical names (Lieberman et al., 2010). The simplest approach is to extract single and consecutive words from a text and then match them to a comprehensive set of toponyms (i.e.,
geographical locations; Schulz et al., 2013). Such a pre-existing list of toponyms is known as a "gazetteer." This approach yields a list of candidate locations independent of the language used in the tweet. The use of a comprehensive gazetteer makes it likely for the algorithm to find locations mentioned in a tweet. Unfortunately, since many location names also have other meanings in normal language usage (e.g., "Darwin" is both a place name and a family name), the results also include many erroneous matches. In contrast, named-entity recognition (NER) analyzes (through natural language processing) the structure
and grammar of the tweet's language (Al-Rfou et al., 2015). Employing NER can help to distinguish, for example, among similarly named place and persons (Amitay et al., 2004). These tools have mostly been developed and trained using more formal texts, such as newspapers (Sultanik & Fink 2012). Nonetheless, researchers have developed several NER approaches for Twitter (Li et al., 2012), most of which are designed for English-language tweets. However, the short, error-prone, multi-lingual nature of tweets, along with that medium's frequent use of slang and abbreviations, has limited the applicability of
NER (Li et al., 2012).

The second step involved in geotagging tweets is *toponym resolution*, which entails assigning a toponym to a specific geographical location (Lieberman et al., 2010). This step is required, because many place names have multiple occurrences worldwide (Leidner, 2007). Most studies have restricted their gazetteers to only include unambiguous place names with a



relatively high population or assigned tweets to the candidate location with the highest population (Amitay et al., 2004). Unfortunately, both approaches introduce errors when an event occurs in a town with both a low population and a name shared with another location. These errors arise because either the town is not included in the limited gazetteer or a city with larger population takes precedence.

For Twitter in particular, challenges persist regarding the automated geotagging analysis of text and other metadata. For example, users rarely post an unambiguous name or a combination of a place and country name, mainly because of the limited length of a tweet (Sultanik & Fink, 2012). Several studies have addressed this issue by using the tweet's metadata as additional spatial information, with examples including the user's hometown (Hecht et al., 2011) and the relationships between users

(Takhteyev et al., 2012). Unfortunately, in many cases, these additional spatial indicators are unavailable or unreliable. Therefore, Schulz et al. (2013) analyzed several spatial indicators, such as the time zone, the user location field, and other textual clues, to obtain a more reliable estimate of a particular tweet's location. Their results revealed that tweet geotagging outcomes can be improved using this method, but only for those tweets with available spatial indicators. As additional spatial information is not always available, this approach cannot be easily applied to all tweets. Moreover, even when this data is

available it does not always match the location mentioned by the user.

This paper outlines the development of a new algorithm, the toponym-based algorithm for grouped geotagging of social media (TAGGS). The algorithm uses grouped geotagging to geotag a much larger percentage of tweets than does the standard approach of individual geotagging. The TAGGS approach has two novel aspects: (1) It permits spatial information from related

tweets to be incorporated in the analysis, allowing users to geotag tweets with few or no spatial indicators of their own. (2) It can geotag tweets at multiple spatial scales (i.e., countries; first- and second-order subdivisions; and cities, towns and villages). We applied TAGGS to over two years of globally sourced tweets with flood-related keywords, collected between July 29, 2014 and February 23, 2017, to demonstrate its use and validate the algorithm using a set of manually geotagged tweets.

**2 Methodology**

The TAGGS algorithm uses geotagging to match a tweet's location references to one or more geographic locations. To that end, a database containing known geo-locations (a "gazetteer") was used to match a tweet's text to one or more candidate locations (*toponym recognition*). Thereafter, additional spatial information obtained from both the tweet itself and related tweets was employed to determine the actual location(s) that the user had mentioned in the tweet (*toponym resolution*). In this chapter, the collection of the input dataset is described (Sect. 2.1). Afterwards, the process of geotagging via toponym

recognition and resolution is outlined (Sect. 2.2).





## 2.1 Input data

The TAGGS algorithm uses three types of input data: a gazetteer, tweets collected using the Twitter API, and additional GIS-based geographical information. To build our gazetteer, we used the GeoNames database (Wick, 2011), a geographical database containing over 4 million cities, towns, villages, and administrative divisions. GeoNames' main dataset contains

towns and villages, including their administrative parent area, geographical location, and population. Another dataset lists alternative names, like translations, slang terms, and abbreviations (e.g., for "New York," it includes, for example, *New York*, *The Big Apple*, *NY*, *Nueva York*), and the language of each alternative name.

The tweets and their associated metadata (e.g., the user's hometown, the user's time zone, and the GPS coordinates of the

device from which the tweet was sent) were collected in real-time via the Twitter streaming API using a series of keywords in 12 major languages, covering a considerable part of the globe (Fig. 1 / Table 1). We collected 35.1 million tweets, posted between July 29, 2014 and February 23, 2017. In addition, we used GIS shapefiles of the global time zones (Muller, 2016) to match user time zones to locations and country and administrative boundaries (Patterson & Kelso, 2017).

## 2.2 Geotagging

Figure 2 describes the procedure followed by the new TAGGS algorithm. First, we collected tweets over a 24-hour period. Each tweet from this timeframe was analyzed on an individual basis (Sect. 2.2.1) by matching its text to our gazetteer (toponym recognition). Next, each of the tweets' candidate locations was assigned a score indicating how well it matched the tweet's additional spatial information (Sect. 2.2.2). While previous approaches have relied on the spatial information of the individual tweet in question, we grouped all tweets according to the toponym (Sect. 2.2.3) identified during the toponym recognition step.

Then, we computed the total score for each candidate location by summing the scores of the individual tweets and using a voting process to assign the best location (toponym resolution) to all tweets in the group (Sect. 2.2.4 / Fig. 3). In addition, a toponym resolution table was made to store the toponyms and their resolved geographic locations of the last time step. This table is later used to geotag tweets in real time. Once locations had been assigned to the tweets, the same procedure was applied to a later timeframe (Sect. 2.2.5 / Fig. 4), which included new incoming tweets. At that stage, tweets older than 24 hours were

no longer considered. Meanwhile, new incoming tweets were immediately geotagged using the toponym resolution table.

### 2.2.1 Toponym recognition

To identify candidate locations for a tweet, a tweet's text was matched to the gazetteer. Tweets are often written in informal language and contain content unnecessary for geotagging. Therefore, we applied Dittrich's (2016) approach to delete URLs and punctuation, split words with medial capitals (camelCase) or underscores, and convert all text to lowercase. Then, all

contiguous sequences of one and two words were extracted from the text ("uni- and bi-grams"). Subsequently, we looked up





all uni- and bi-grams in our gazetteer, and the result was a list of potential candidate locations for each toponym mentioned in the text. We then further filtered the results to obtain the candidate locations, using the following approach:

1.  All uni- and bi-grams among the 1,000 most common words (Pishdar, 2014) in a tweet's language are discarded.

2.  Locations are frequently referenced to using alternative names in other languages (e.g., "New York" is "Nueva York" in Spanish). If a tweet referenced a location via an alternative name, we only considered that tweet if it uses the same language as that alternative name (e.g., an alternative name for "Cameroon" is "Cameron," meaning that all mentions of Cameron, the UK's former prime minister, would otherwise be located in the "Cameroon").

3.  Since locations with small populations were not frequently mentioned in tweets, we followed Amitay et al. (2004), and discarded those locations with a population of less than 5,000 persons.

4.  Next, all locations with names that were part of another location's name are discarded. For example, a tweet containing New York matches both York and New York, although the user was clearly referring to New York. Therefore, we excluded York.

5.  If a tweet mentioned a location with a name simultaneously used for a province or country and an identically named town within that area (e.g., the city of New York is within the state of New York), the location with the most translations provided in the gazetteer—a criterion that we used as a proxy for importance—took precedence (i.e., the city of New York took precedence over the state).

### 2.2.2 Scoring

For each tweet for which we found one or more candidate locations, as described in Sect. 2.2.1, its additional spatial indicators were matched to each candidate location. The bulleted list below describes the matching process for each of the spatial indicators.

- User time zone: Twitter's time zone field signifies an area with a uniform time standard. Twitter initially sets users' time zones, but users can manually adjust this setting. Our gazetteer contained a list of time zones for each location, used to match these to the user's time zone.

- Coordinate-based indicators: We extracted geographical coordinates for two spatial indicators (see below). We considered the coordinates extracted from these indicators a match for a candidate location if they were located within 200 km of each other or, for administrative areas, if the coordinates were within the same country as the candidate location.

  o User hometown: Users can specify their hometowns in their user profile. In doing so, users receive assistance from a dynamic menu of location options that appears when they start typing in the Twitter text field. Although the box can be ignored, most users do make use of it. This means that in most cases, the location field is either (1) a town and country name separated by a comma or (2) a country name. However, many variations are possible, including fantasy places (Schulz et al., 2013), multiple locations, and incomplete data entries (e.g., a user who lives in Washington, D.C. might simply enter



"Washington" in the location field). We searched for both the town and country in the gazetteer to create a list of candidate towns within the specified country. If no comma was present, we looked up the entire field in the gazetteer.

- o Location: When a tweet is sent from a GPS-enabled device, and when the user's privacy settings or manual adjustments assign a location to the tweet, that user's location at the time of posting is attached to the tweet. Additionally, the user can attach a geographic entity to a tweet, by manually selecting it from a dynamic list.

- Mentions of related places: We matched user mentions of other locations higher (geographical parent) or lower (geographical child) in the hierarchy (e.g., "Los Angeles" is the geographical child of "California," which is the geographical child of "the United States"), other towns within 200 km of a candidate place name, and other administrative areas within the same geographical parent (Amitay et al., 2004).

Next, we used a scoring system to indicate the likelihood of a match between the location referenced in the tweet and each candidate location. An overview of the scores for each of the five spatial indicators is provided in Table 2. These scores were summed to obtain the total score (maximum of 7), which indicated the likelihood of a match.

### 2.2.3 Grouping

We assumed that multiple tweets that mentioned the same toponym within a given timeframe referred to the same location (e.g., if a flood occurred in Boston, UK, we expected that all users mentioning "*flood*" and *"Boston"* were referencing Boston, UK, rather than Boston, Massachusetts, US.). All tweets mentioning the same toponym were then grouped together. Thus, the greater the number of tweets mentioning a location, the larger was the associated group—and therefore, the higher the probability of metadata being available for that group. Since tweets could contain multiple toponyms, individual tweets could belong to more than one group.

### 2.2.4 Voting and assigning locations

In this step, for each group, the total score of each candidate location was computed by averaging the scores for the candidate locations of the individual tweets (Fig. 3). If multiple tweets originated from the same user and thus had the same metadata, only the most recent tweet was considered. In addition, because the mentions of related places rely on textual clues, and since users frequently copy each other's tweets, we only considered the oldest tweet for clusters of similar tweets. To that end, we created word vectors for the tweets within a group and then compared those vectors. If the vectors were similar, we eliminated the newest tweet. For further details on this approach, refer to Hürriyetoğlu et al., 2016.

Finally, we assigned the location with the highest score to all tweets in the group if that tweet's referenced toponym was the official name of the location or if the tweet's language matched the toponym language. If multiple locations had an equally




high score, we assumed that the correct location was the candidate with the highest population. Tweets that were geotagged in multiple locations are discarded if they were further than 1,000 km apart (e.g., "*Amsterdam Weather – Floods in Argentina*"), unless these locations were both a country or a continent (e.g., '*Whoah! Floods in the UK and USA!!*').

Moreover, it was also possible to discard potential locations for which the average score was below a certain threshold. When no minimum score (i.e., a minimum score of 0) was set, a large number of tweets was assigned to incorrect locations, due to a lack of matching metadata (e.g., numerous tweets were assigned to the city of "Mobile" in Alabama, US.). By increasing the threshold to, for example, 0.2, groups with little to no metadata matching any of the candidate locations were discarded. This meant that the recall decreased (i.e., fewer tweets were assigned locations), while the precision of the algorithm increased.
Introducing a much higher threshold, such as 1.0, would have improve precision but also would have meant discarding a much higher percentage of tweets. Therefore, we decided to initially set a 0.2 threshold and to perform a sensitivity analysis (Sect. 3.2).

In addition, the toponyms and their respective resolved locations were saved in a *toponym resolution* table. That table indicated
the location with the highest score per toponym, and therefore, the location most likely for a future tweet to reference. This toponym resolution table was continuously updated and used to geotag new incoming tweets.

### 2.2.5 Iteration

To continuously geotag tweets, we used an iterative process (Fig. 4). When the geotagging of all tweets posted during the 24 hour time window, the time window was moved. All new tweets posted during the previous iteration was running were
retrieved from the tweet database and separately analyzed for toponyms and respective spatial indicators (Sect.s 3.3 and 3.4), while tweets older than 24 hours were excluded. The locations mentioned in those tweets were, again, resolved (Sect. 3.5) and used to update the *toponym resolution table*. As the tweet geotagging process included information from other tweets (including locations referred to in future tweets), it was possible for a tweet's location and respective score to change. In such cases, we updated the database accordingly. Such alterations only occurred when in a subsequent iteration, we found a higher
score for a specific location or identified another location with a higher score (but the same toponym). Because a 24-hour time interval was used when processing the tweets, those tweets older than 24 hours did not influence new tweets' scores during the updating process.

In addition, when the first iteration was completed, another process analyzing incoming tweets in real-time was initiated. Using
the procedure described in Sect. 2.1, the text of the tweets was processed, and its uni- and bi-grams are matched with the *toponym resolution table*. This resulted in an initial guess regarding the locations mentioned in each tweet.



## 3 Results

### 3.1 Application of TAGGS

We applied TAGGS on the 35.1 million tweets that we collected. Although the algorithm was designed for real-time usage, we applied it to a historical dataset and ran it as if data were available in real-time. We therefore iterated over the data with a

24-hour time window, shifting the time window by 6 hours in each step. To increase the accuracy of the algorithm, a certain threshold was used (see Sect. 2.2.2), so that all locations found in tweets that scored below the threshold (Sect. 2.2.4) were discarded. The results for a 0.2 threshold are summarized in Table 3, and the results of a sensitivity analysis for the threshold value are presented in Sect. 3.2.3.

Of the 35.1 million tweets, we found that 11.6 million mentioned at least one location, and 1.9 million tweets referenced multiple locations. In addition, when distinguishing between administrative levels, roughly half of the locations referenced the city, town, or village level, while a quarter of the locations cited both the country and the lower administrative level, respectively.

To gain insight into the geotagged tweets, those countries covered by the algorithm (Fig. 1) with a population of at least 10 million people were grouped according to economic development. For that purpose, we employed the income groups defined by the World Bank (The World Bank, 2017). For each group, the number of geotagged tweets between August 2014 and December 2016  was plotted against the total flood losses over this period, as described in the Munich Re database (Munich Re, 2016) on a purchasing power parity (PPP) basis (The World Bank, 2016; Fig. 5). This gives an impression of how many

tweets were found for each country and how Twitter reporting relates to flood impacts. The data made clear that in high-income (green) countries, there were about one to two orders of magnitude more tweets than in low-income (orange) countries. The number of tweets in middle-income (blue & yellow) countries fell between the other two groups, with a particularly large spread in the lower-middle-income (yellow) countries. Notably, these numbers likely reflect a size effect, as Indonesia (IDN) and India (IND), which had the highest number of tweets within the lower-middle-income group, also have large populations.

However, the results underscored that relatively small countries, such as Malaysia (MYS) and Nepal (NPL), generated a significant number of (geotagged) flood tweets within their respective groups. These findings suggest that flood events, and not just the size of the population or the Twitter user base, are responsible for the high number of tweets during the investigated time period.

The plots also illustrate that in general, more flood tweets seemed to be linked to higher levels of flood damage over the study period, as the points roughly go from bottom left-hand corner to the top right-hand corner of the diagrams. This relation is influenced by many other factors, including (but not limited to) variations in the extent of Twitter usage per country, language use per country, and keyword selection, and is therefore by no means strong enough to have any predictive power after



regression analysis. That said, the existence of this relationship was in line with expectations. Namely, in countries that suffered from disastrous flood events that caused significant damage, a substantial number of tweets about flooding were generated. This finding indicated that the algorithm seemed to be successful in capturing flood events around the globe.

## 3.2 Validation of TAGGS

To properly validate the TAGGS algorithm, we defined a baseline with *manually* geotagged tweets. To achieve that goal, we adopted an approach similar to that of a human reading tweets and extracting the locations from it. Thus, we only recognized locations in instances in which it could be reasonably assumed that a human could resolve the location (e.g., "a flood in block five," in the absence of other spatial indicators, would not be linked to a particular known location). However, we did include, for example, adjectives and other terms referencing locations that an algorithm might not recognize as such (e.g., "the English

weather" would be recognized by a human as referring to England, whereas an algorithm would generally not resolve this). When multiple locations were mentioned in a tweet, we manually assessed whether they represented alternate means of citing a single location or distinct locations. When two locations appeared to be referring to the same place, we tagged them as a single location (e.g., for "there is a flood at the A10 near Amsterdam," we recognized both "Amsterdam" and "A10" as references to a flood in Amsterdam). On the other hand, if we recognized multiple, distinct locations (e.g., "flooding in United

States and England"), we tagged the tweet accordingly.

We manually tagged a random sample of tweets on two separate days:

- Dec 12, 2015: To check if our model properly for small flood events in multiple languages, we selected a day during which multiple such events occurred across the globe, including in Indonesia, India, Kenya, Congo, Norway, the UK, Canada, and Paraguay (1,581 tweets)/

- Dec 27, 2015: When the number of tweets that mentioned a specific location higher, the probability of sufficient metadata being available is also higher. Therefore, we validated our algorithm on a date with multiple large events. On the date in question, several major floods received global news coverage, including floods in the US, the UK, and Argentina (1,355 tweets).

Then, we compared the manually tagged tweets to both the automated individual and automated grouped geotagging (TAGGS)

approaches. For individual geotagging, we use the location metadata but did not consider other tweets mentioning the same geographical entities, similar to Schulz et al. (2013). Both the grouped and individual automated geotagging procedures were performed with two thresholds (0 and 0.2).

### 3.2.2 Number of locations found

Figure 6 illustrates the number of locations identified using the different approaches and the number of erroneous matches.

Using the manual approach, of the 2,936 total tweets in our validation set, we found 2,190 locations in 1,657 tweets, because some tweets mentioned multiple locations. Using individual geotagging we found approximately 29% of these locations, of





which roughly 87% were correct. The grouped geotagging technique developed in this paper increased the number of found locations to approximately 59%, of which about 89% are correct. In contrast, of the 2,936 tweets, only 58 (~2%) have coordinate information attached (Fig. 6). This suggests that the TAGGS approach makes significantly more spatial information available than does a strategy relying on either individual geotagging or coordinates alone. Thus, the results indicate the

feasibility of this new approach.

### 3.2.3 Precision versus recall: varying the threshold

With geotagging algorithms, there is a trade-off between the number of tweets that are tagged (recall) and the number of correctly tagged tweets (precision; Leidner, 2007). Precision measures assess the number of correctly geotagged tweets relative to the total number of geotagged tweets. Hence, precision markers do not provide an indication of the total number of tweets

within a location. Recall measures reflect the number of correctly geotagged tweets relative to the total number of tweets with a spatial reference. Basically, the greater the level of precision (i.e., the smaller the number of incorrect tags), the smaller the total number of geotagged tweets. Inversely, if one wants to geotag more tweets (higher recall), the number of errors within the geotagged tweets (in terms of incorrect location assignments) will also increase (lower precision).

The trade-off between precision and recall can be clearly seen in Tables 4 and 5, which display the percentages of correct and incorrect location matches. Those tables demonstrate that the number of tweets without geotagged locations was lower using a threshold of 0 than a threshold of 0.2. However, at the same time, with the 0.2 threshold, the number of errors (falsely or incorrectly geotagged) decreased considerably, implying that precision increased with a higher threshold. The results demonstrated that this was particularly the case for the new grouped algorithm. With that approach, the number of incorrectly

geotagged tweets decreased, and the number of falsely geotagged tweets (i.e., tweets that should not have had a location but that were given one) fell dramatically (13.1% to 5.1% and 14.6% to 6.6% of all tweets for December 12, 2015 and December 27, 2015 respectively).

With individual automated geotagging, we found that a higher level of precision was achieved at the expense of the total

number of geotagged tweets (67.1% and 56.7% of tweets were not geotagged using the 0.2 threshold, instead of 27.5% and 21.9% for December 12, 2015 and December 27, 2015 respectively). This resulted in only 27.6-38.2% of correctly geotagged tweets. Hence, the trade-off between precision and recall remained very strong when using the individual geotagging algorithm.

When using the newly developed grouped algorithm, this trade-off still existed, but it had a much less pronounced effect. Instead, the number of tweets that were not geotagged increased only mildly, whilst the number of falsely and incorrectly geotagged tweets declined considerably. As a result, a very high percentage (64.7% and 74.1%) of tweets were correctly geotagged.





### 3.2.4 Effect of the event size

Comparing Tables 4 and 5 highlights differences in performance due to different flooding circumstances. On December 12, 2015, there were various smaller flood events, whilst on December 27, 2015, a couple of very large flood events took place. These two cases make clear that the algorithms (both individual and grouped) were approximately 10% more accurate for larger-scale flood events than smaller-scale flood events. Such a finding is to be expected, because during the large flood events in the US and UK, both countries with a high level of Twitter usage, a larger percentage of tweets mentioned the same toponym. The grouping approach meant most of these tweets were scored, even though not all them had spatial information available. In contrast, single tweets without location metadata were not geotagged. This latter situation is more common when a higher number of smaller events occur. Nevertheless, the grouped algorithm still correctly geotagged about two-thirds of the tweets with a location, even on days with predominantly smaller flood events.

### 4 Concluding remarks and outlook

In this paper, we presented TAGGS, a multi-lingual algorithm that groups topologically related tweets based on their referenced toponyms and then geotags those tweets using the mutual spatial information of the entire group. In addition, the algorithm successfully differentiates between various administrative levels.

Studies on event detection often work with geo-located tweets by using the coordinates attached to them. In our validation set, however, only 2% of all tweets actually had coordinates attached. By geotagging tweets using the tweet content, this study doubled the number of correctly geo-located tweets when a 0.2 threshold was employed. Moreover, using the grouping approach developed in TAGGS also boosted the precision level without lowering the number of geotagged tweets (i.e., lowering recall) to an unacceptable degree. As a result, approximately 65-75% of tweets were correctly geotagged. At the same time, the number of incorrectly geotagged tweets remained below 10% for all tweets with a location, while the number of tweets that were falsely given a geotag was low, at only 6% (of all tweets).

Unfortunately, our algorithm also introduced several minor problems: (1) Using the individual geotagging approach, a tweet is only tagged in a location if the metadata matches that location. When a tweet mentions a location with a toponym that is also frequently used in normal speech, all tweets mentioning this word can be tagged in that location, rather than only those tweets that used the word as a toponym. An example of this is "turkey," a term that can refer to both the country of Turkey and the bird of the same name. (2) In rare cases, when a flood occurs in two different places with identical place names, all tweets are put into one group and hence tagged in in only one of these locations. (3) Tweets often mentioned areas (e.g., the East Coast), rivers, and airports. Although the algorithm can resolve such locations using metadata, many such areas have not been included in this study's gazetteer. Including these entities in the gazetteer could improve the recall of the algorithm.



Currently, using the approach described in this paper, we only tag each tweet using the spatial information from that tweet itself and from other tweets mentioning the same toponym. In future research, we plan to expand on this approach by detecting sudden changes in the number of mentioned locations in an area. This technique would allow us to improve the geotagging algorithm by taking into account sudden increases in mentions of nearby locations, using such a rise as an additional spatial indicator. In addition, it would allow us to create a detection and monitoring algorithm for disaster events based on sudden changes in the number of tweets. Moreover, while this paper focused on tweets, the approach outlined within this text has other applications. In addition, this method could be combined with other types of mass data, such as newspapers and other social media platforms, to yield even more geotagged information.

## 5 Code availability

The algorithm uses Python 3 scripts and connects to an Elasticsearch and PostgreSQL (with PostGIS) database. All code is publicly available on GitHub (https://github.com/jensdebruijn/TAGGS) as well as Zenodo (http://doi.org/10.5281/zenodo.802960).

## 6 Data availability

For this research, tweets publicly available from the Twitter streaming API are used as well as other open source data. Information about this input data is provided in the aforementioned GitHub repository. The authors are willing to share selected geotagged tweets to other researchers in line with Twitter's privacy policy.

## 8 Author contribution

Jens de Bruijn, Hans de Moel and Jurjen Wagemaker designed the algorithm. Jens de Bruijn coded the algorithm and writing was done by Jens de Bruijn, Hans de Moel, Brenden Jongman and Jeroen Aerts.

## 9 Competing interest

The authors declare that they have no conflict of interest.

## 10 Acknowledgements

The research leading to these results has received funding from Netherlands Organisation for Scientific Research (NWO) VICI (grant number 453-14-006) and the European Community's Seventh Framework Programme (FP7) ENHANCE (grant number 308438).





## 11 Tables

Table 1: Keywords related to floods, and percentage of tweets per language over the period 2014-2017.

| Language | Keywords | Number of tweets per language (%) |
|---|---|---|
| English | flood, floods, flooding, flooded, inundation, inundations, inundated | 59.87 % |
| Indonesian | banjir, banjirjkt, bantubanjir | 17.88 % |
| Filipino | baha, bumabaha, apaw, pagbaha, pag-apaw, guho, koppu, typhoonkoppu | 2.13 % |
| French | inonder, inondation, submerger, noyer, engorger, avalanche, maree haute, torrent de l'eau | 0.65 % |
| German | flut, hochwasser, sintflut, Überflutung, erdrutsch | 0.06 % |
| Italian | inondazione, inondacioni, frana, alluvione, Acqua alta, acquaalta, aquaalta, passerelle venezia, sotto acqua, sott'acqua, Paratia, paratie | 0.19 % |
| Dutch | overstroming | 0.04 % |
| Polish | powódź, powodzie, potop, przypływ, wylew rzeki, zalanie, zalew, zalwewie, zatopienie, osuwisko | 0.03 % |
| Serbian | poplava, poplave, поплава, поплаве, klizišta, Ландслиде | 0.05 % |
| Portuguese | inundação, inundacão, inundaçao, inundacao, alagar, transbordar, jorrar, diluvio, inundações, deslizamento de terra | 0.69 % |
| Spanish | inundación, inundacion, inundar, torrente, desbordar, anegar, diluvio, pleamar, inundaciones, deslizamiento de tierra, deslizamiento tierra | 11.45 % |
| Turkish | sel, taşkın, tufan, su baskını | 6.96 % |



Table 2: : Score for each of the spatial indicators assigned to individual tweets. The scores are in the order of magnitude found by Schulz et al. (2013).

| Indicator | Score |
|---|---|
| **Time zone** | 1 |
| **User home town** | 1 |
| **Coordinates** | 2 |
| **Mentions of related places** | 3 |

Table 3: Results of the automated geotagging of 35.1 million tweets (using a threshold of 0.2; see Sect. 2.2.4)

| | Number of tweets |
|---|---|
| Total | 35.1 million |
| One or more location | 11.6 million |
| Multiple locations | 1.9 million |
| Country level | 2.6 million |
| Lower administrative level | 3.3 million |
| City, town, village etc. | 6.9 million |

Table 4: Results of the TAGGS validation using 1,355 tweets posted on December 12, 2015, with thresholds of 0 and 0.2.

| 12 December 2015 | Individual (%) | | Grouped (%) | |
|---|---|---|---|---|
| | *thresh. 0* | *thresh. 0.2* | *thresh. 0* | *thresh. 0.2* |
| Falsely geotagged (should not have location) * | 18.1 | 18.1 | 13.1 | 5.1 |
| Incorrectly geotagged** | 10.8 | 5.3 | 11.2 | 8.2 |
| Not geotagged** | 27.5 | 67.1 | 21.9 | 27.1 |
| Correctly geotagged** | 61.8 | 27.6 | 66.9 | 64.7 |

* percentage of all tweets

** percentage of all tweets with location




Table 5: Results of the TAGGS validation using 1,581 tweets posted on December 27, 2015, with thresholds of 0 and 0.2.

| 27 December 2015 | Individual (%) | | Grouped (%) | |
|---|---|---|---|---|
| | *thresh. 0* | *thresh. 0.2* | *thresh. 0* | *thresh. 0.2* |
| Falsely geotagged (should not have location) * | 22.6 | 22.6 | 14.6 | 6.6 |
| Incorrectly geotagged** | 15.3 | 5.1 | 11.1 | 9.2 |
| Not geotagged** | 21.9 | 56.7 | 14.6 | 16.7 |
| Correctly geotagged** | 62.8 | 38.2 | 74.3 | 74.1 |

* percentage of all tweets

** percentage of all tweets with location

**11 Figures**

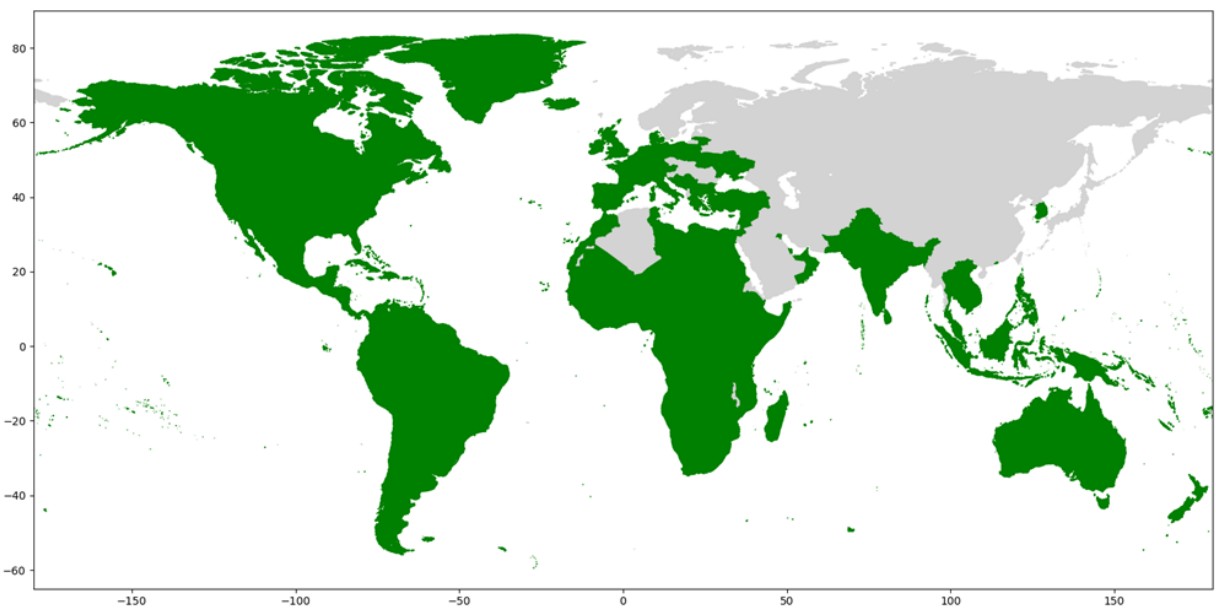

**Figure 1: Countries where at least one of the languages spoken, as specified in the GeoNames (Wick, 2011) database, is analyzed.**





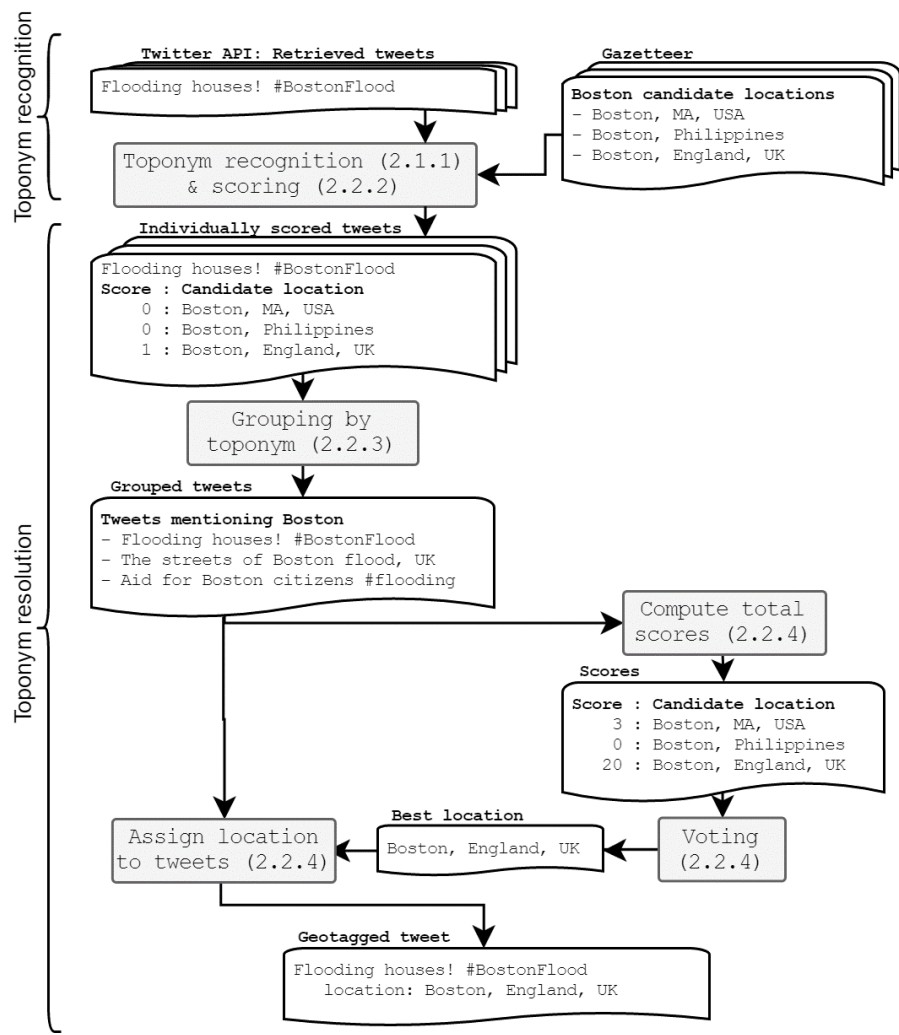

**Figure 2: Overview of the TAGGS geotagging process.**

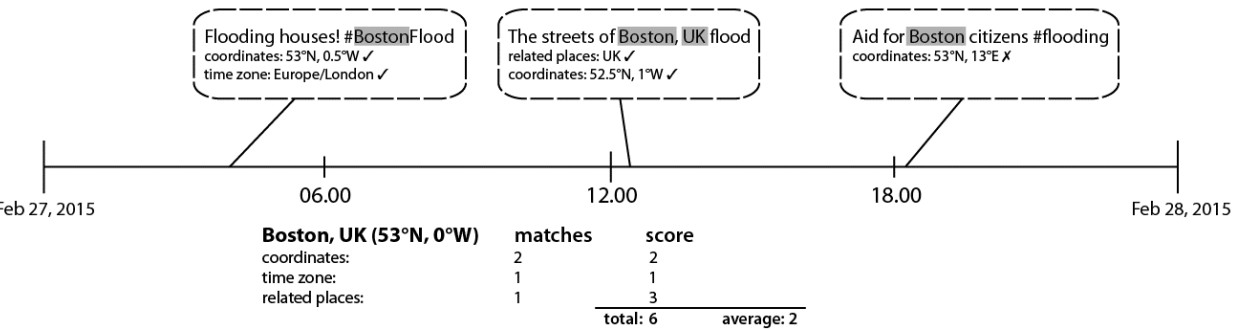

**Figure 3: In this example, three tweets mention Boston within 24 hours. The metadata of the first and second tweet match Boston, UK, while the third one did not have any matching spatial indicators. Using the spatial information for the first two tweets makes it**





possible to correctly assign the third tweet to Boston, UK. In this example, the total score for Boston, UK, is 6 and the average score is 2.

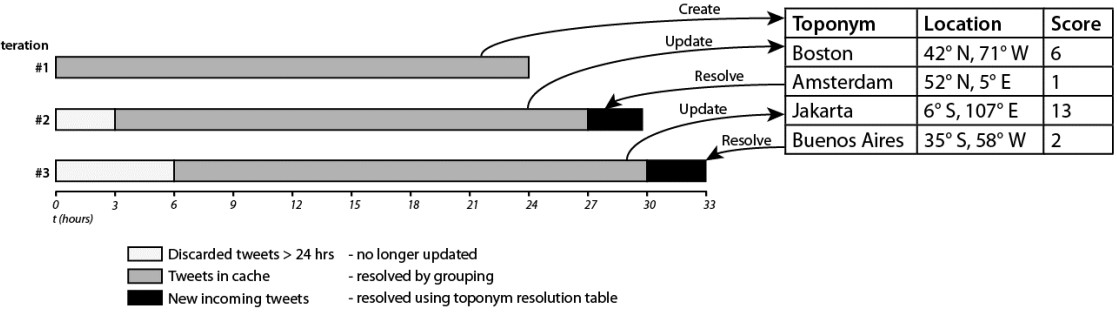

**Figure 4: Schematic overview of the geotagging process. As indicated, 24 hours' worth of cached tweets were geotagged by grouping and analyzing their additional spatial information. Moreover, this step entailed creating or updating the toponym resolution table used to geotag subsequent incoming tweets. Tweets older than 24 hours were excluded from this process.**





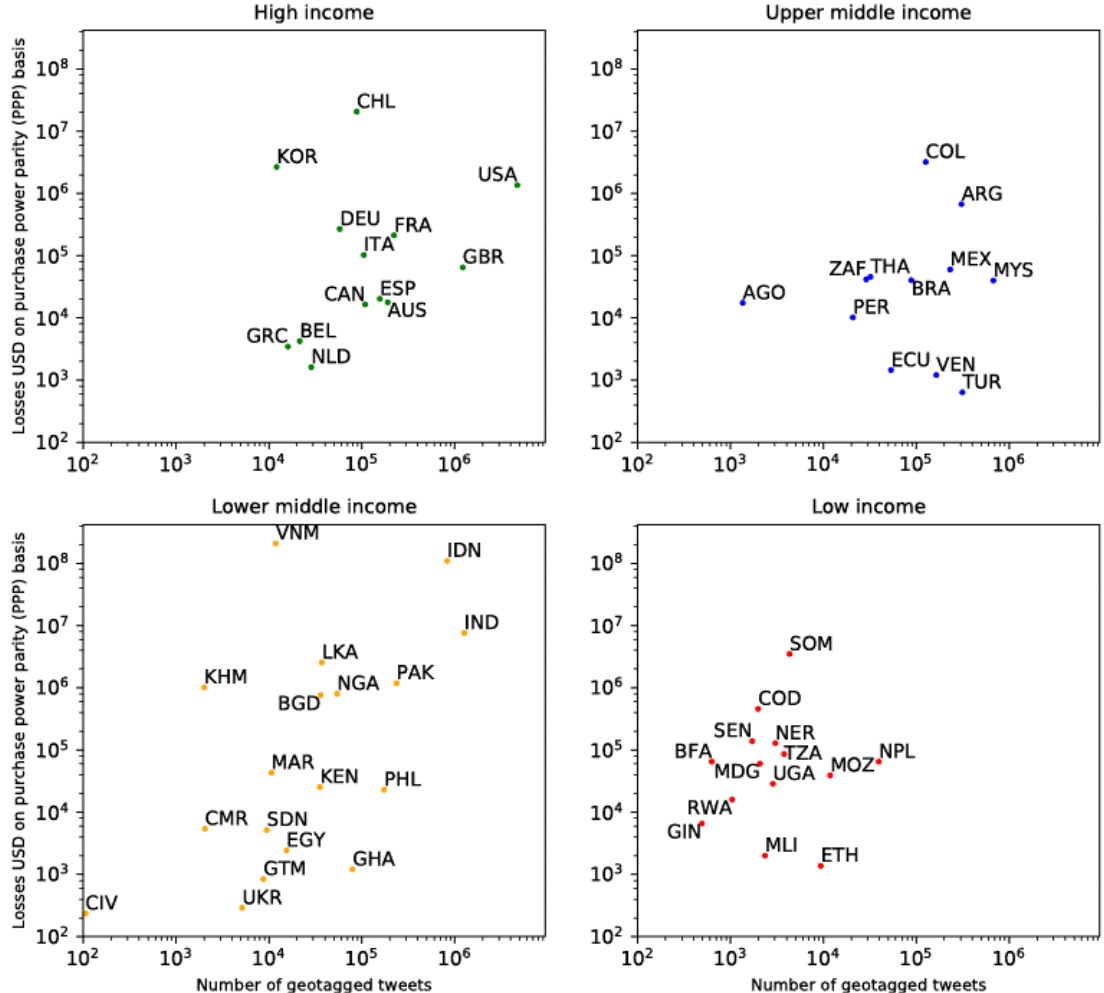

**Figure 5: The number of geotagged tweets relative to losses due to flood events between July 29, 2014 and December 31, 2016 for four country income groups.**





| | Number of locations | |
|---|---|---|
| | Total | Correct (%) |
| Manual | 2190 | 100 |
| Coordinates | 58 | 100 |
| Individual geotagging | 641 | 87 |
| Grouped geotagging | 1302 | 89 |

**Figure 6: Comparison of the number of geo-located tweets our the validation set in the middle of the UK for various geo-location methods. The green dots represent correctly identified locations, and the red dots represent incorrectly identified locations.**

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
