# Peer review of "TAGGS: Grouping Tweets to Improve Global Geotagging for Disaster Response"

_Natural Hazards and Earth System Sciences, 2017_

## Referee Comment (RC1) · Anonymous Referee #1 · 19 Jun 2017

This paper presents TAGGS, an innovative method to group natural hazards related Twitter tweets, which is very useful for the response and rescue after the natural hazards happen, mitigating the loss. Overall, this paper fits the interest of NHESS Journal; given the high-quality of its scientific innovation and writing, the paper deserves an acceptance, though some minor revisions are needed.

The paper puts emphasis on its innovative geotagging algorithm, namely TAGGS, which basically is a method dealing with toponym recognition and resolution, especially for tweets. Although it does a great job reviewing related works, it overlooks some toponym recognition and resolution work on short texts, which could be useful for the case of tweets as well. Moreover, only those fields in the meta-data are considered as spatial indicators. What about the context in the tweet itself? For example,

if a tweet mentions "Washington" and "president", it is very likely the "Washington" is referring to Washington D.C.. This could be the next step if the authors are going to further their approach. Here are two related literature that the authors may refer:

Ju, Y., Adams, B., Janowicz, K., Hu, Y., Yan, B., & McKenzie, G. (2016). Things and Strings: Improving Place Name Disambiguation from Short Texts by Combining Entity Co-Occurrence with Topic Modeling. In Knowledge Engineering and Knowledge Management: 20th International Conference, EKAW 2016, Bologna, Italy, November 19-23, 2016, Proceedings 20 (pp. 353-367). Springer International Publishing.

Y Hu, K Janowicz, S Prasad (2014): Improving Wikipedia-based place name disambiguation in short texts using structured data from DBpedia, In Proceedings of 8th ACM SIGSPATIAL Workshop on Geographic Information Retrieval, Nov. 4, 2014, Dallas, TX, USA.

Some parts of the writing could be clarified or improved:

In line 24 of Section 2.2, the expression "tweets older than 24 hours" is confusing. Also, what is the reason to choose "24 hours" as the scanning window? What's the difference if I choose "6 hours" or "72 hours"?

It is nice to see "thresholds" are used to balance between precision and recall, but it seems like the authors only use "0" and "0.2". It would be better to see a precision-recall curve, which is typical for the task of information retrieval.

In figure 3, for Toponym recognition, it should be 2.2.1, instead of 2.2.1.

---

## Referee Comment (RC2) · Anonymous Referee #2 · 20 Jun 2017

Geoparsing and Geotagging is a well studied problem. The authors have missed a lot of important work in related work section (see end citations).

Although the problem is based on disaster response actually the work presented is standard geoparsing.

The location identification approach proposed by the authors is not state of the art (standard tokenization, geonames gazetteer lookup).

The location disambiguation is also quite simplistic (language/time zone/population filters, location name subsumption). Modern geoparsing approaches use extra information to boost precision, either from contextual text (e.g. linguistic patterns used and other location mentions in text), geospatial source (e.g. OpenStreetMap database locations for spatial proximity) or social context (e.g. social media tags from a training corpus). The approach used by the authors is not advancing the state of the art.

The idea of clustering (i.e. grouping) posts with the same location mention to provide additional context is potentially a novel idea that does advance the state of the art.

However the choice to filter out all locations where the population density is small greatly weakens this works applied value. Geoparsing popular locations, usually cities with a high population density, is much easier than geoparsing obscure remote locations. For populous areas social media tagging has been proven to be very effective. Such work (see Kordopatis-Zilos) has not been cited at all and is not compared.

There are publicly available benchmark geoparse datasets (see PyPI geoparsepy https://pypi.python.org/pypi/geoparsepy) which could and should have been used to directly compare results to published work (see Middleton and Gelernter). The authors instead created a smaller dataset manually and thus the results are not directly comparable.

The evaluation itself is weak. No statistical significance is reported. A breakdown of results by event & language is not provided (known factors in performance). The authors appear to be geoparsing at the region level (the paper is not precise about this so its impossible to tell). The example of 'A10 near Amsterdam' should geoparse both the A10 (road) and Amsterdam (region) - but appears to only score on Amsterdam. The whole approach lacks rigour and depth.

The results themselves look weak (i.e precision scores), which is not surprising considering the lack of state of the art location disambiguation techniques. I would expect a precision for region level geoparsing to be 0.9+ (F1 0.8+). The authors manage around 0.7 precision.

Overall this work is too immature at this stage for a journal publication. The lack of a real comparison to the extensive related work already published in this area means

this work has little current value to a serious researcher.

– missing citations –

Judith Gelernter and Nikolai Mushegian, 2011. Geo-parsing Messages from Microtext. Transactions in GIS, Vol 15, Issue 6, 753–773

Middleton, S.E. Middleton, L. Modafferi, S. "Real-time Crisis Mapping of Natural Disasters using Social Media", Intelligent Systems, IEEE , vol.29, no.2, pp.9,17, Mar.-Apr. 2014

Middleton, S.E. Krivcovs, V. "Geoparsing and Geosemantics for Social Media: Spatio-Temporal Grounding of Content Propagating Rumours to support Trust and Veracity Analysis during Breaking News", ACM Transactions on Information Systems (TOIS), 34, 3, Article 16 (April 2016), 26 pages.

Marieke van Erp, Giuseppe Rizzo and Raphaël Troncy, 2013. Learning with the Web: Spotting Named Entities on the intersection of NERD and Machine Learning. In Proceedings of the 3rd Workshop on Making Sense of Microposts (#MSM2013). Rio de Janeiro, Brazi

Alan Ritter, Sam Clark, Mausam and Oren Etzioni, 2011. Named entity recognition in tweets: An experimental study. In Procedings of Empirical Methods for Natural Language Processing (EMNLP), Edinburgh, UK

Alexandre Davis, Adriano Veloso, Altigran S. da Silva, Wagner Meira Jr., Alberto H. F. Laender, 2012. Named entity disambiguation in streaming data. In Proceedings of the 50th Annual Meeting of the Association for Computational Linguistics, 815 - 824, Jeju, Republic of Korea

Giorgos Kordopatis-Zilos, Symeon Papadopoulos, and Yiannis Kompatsiaris. 2015. Geotagging social media content with a refined language modelling approach. In Pacific-Asia Workshop on Intelligence and Security Informatics. Ho Chi Minh City, Vietnam. 19 May 2015

---

## Author Comment (AC1) · 30 Aug 2017

We thank the reviewer for reviewing our study. Below we list the reviewer's comments in bold and discuss how we incorporated them in our paper. This paper presents TAGGS, an innovative method to group natural hazards related Twitter tweets, which is very useful for the response and rescue after the natural hazards happen, mitigating the loss. Overall, this paper fits the interest of NHESS Journal; given the high-quality of its scientific innovation and writing, the paper deserves an acceptance, though some minor revisions are needed.

We thank the reviewer for his/her encouraging words and endorsement of our innovative approach.

[Figure]

The paper puts emphasis on its innovative geotagging algorithm, namely TAGGS, which basically is a method dealing with toponym recognition and resolution, especially for tweets. Although it does a great job reviewing related works, it overlooks some toponym recognition and resolution work on short texts, which could be useful for the case of tweets as well. Moreover, only those fields in the meta-data are considered as spatial indicators. What about the context in the tweet itself? For example, if a tweet mentions "Washington" and "president", it is very likely the "Washington" is referring to Washington D.C.. This could be the next step if the authors are going to further their approach. Here are two related literature that the authors may refer:

Ju, Y., Adams, B., Janowicz, K., Hu, Y., Yan, B., & McKenzie, G. (2016). Things and Strings: Improving Place Name Disambiguation from Short Texts by Combining Entity Co-Occurrence with Topic Modeling. In Knowledge Engineering and Knowledge Management: 20th International Conference, EKAW 2016, Bologna, Italy, November 19-23, 2016, Proceedings 20 (pp. 353-367). Springer International Publishing.

Y Hu, K Janowicz, S Prasad (2014): Improving Wikipedia-based place name disambiguation in short texts using structured data from DBpedia, In Proceedings of 8thA CM SIGSPATIAL Workshop on Geographic Information Retrieval, Nov. 4,2014, Dallas, TX, USA.

We agree with the reviewer that these studies are indeed both interesting and valuable and would help to improve our approach in further research. Unfortunately, this is currently beyond the scope of our work and, therefore, we included both references in the section on future work and possible improvements (Sect. 4), which now reads like this:

In future work, we aim to continue improving our algorithm. Currently, using the approach described in this paper, we only parse each tweet using the spatial information from that tweet itself and from other tweets mentioning the same toponym. In future research, we plan to expand on this approach by detecting sudden changes in the

number of mentioned locations in an area. This technique would allow us to improve the geoparsing algorithm by considering sudden increases in mentions of nearby locations, using such a peak as an additional spatial indicator. Other improvements could be made by taking into account additional context, such as entity co-occurrence (Hu et al., 2014; Ju et al., 2016) or the geography of Twitter networks (Takhteyev et al., 2012).

Some parts of the writing could be clariïñĄed or improved: In line 24 of Section 2.2, the expression "tweets older than 24 hours" is confusing. Also, what is the reason to choose "24 hours" as the scanning window? What's the difference if I choose "6 hours" or "72 hours"?

In line with the reviewer's comments we have revised several sentences:

Once locations had been assigned to the tweets, the same procedure was applied to a later scanning window (Sect. 2.2.5 / Fig. 4), which included new incoming tweets. At that stage, tweets that are outside the scanning window were no longer considered. Meanwhile, new incoming tweets were immediately geoparsed using the toponym resolution table.

and

All new tweets were retrieved from the tweet database and separately analyzed for toponyms and respective spatial indicators (Sect. 2.2.1 and 2.2.2), while tweets that fall outside of the scanning window were discarded.

In addition, we have varied the size of the scanning window and included a detailed analysis in the paper. The text and figures now read as follows:

Figure 1 shows the recall and precision measures for a varying scanning window size, ranging between 6 minutes and 48 hours. In theory, when using an infinitesimally small scanning window for grouped geoparsing, the results would be identical to the individual geoparsing. It is clearly visible that, in general, both precision and recall increase when the size of the scanning window is larger. This is expected, because a larger

number of tweets are grouped and therefore, the likelihood that spatial information is available increases. Although an increase of recall and precision is still visible for a larger scanning window, the increase is not substantial, which indicates that spatial information is available for most toponyms. When new floods occur, it is not feasible to take location mentions of previous floods into account. Therefore, we hypothesize that when the scanning window becomes too large, the performance of the model will be lower. Unfortunately, because of memory (RAM) constraints in our current setup, we cannot test this. Ideally, the size of the scanning window depends on the volatility of the event type, where events with a longer average duration (people will likely refer to the same event over a longer timespan), such as droughts, could benefit from a larger scanning window and vice versa for shorter events.

It is nice to see "thresholds" are used to balance between precision and recall, but it seems like the authors only use "0" and "0.2". It would be better to see a precisionrecall curve, which is typical for the task of information retrieval.

We thank the reviewer for his/her valuable comment and have updated the analysis of the validation study. We now show a precision-recall curve for each administrative level for both singular and grouped geoparsing. The text and figures now read as follows:

Figure 2 shows the recall and precision scores for individual and grouped geoparsing with a varying threshold. The trade-off between precision and recall is visible in the first window: When a higher threshold is chosen, more location matches are discarded, while the likelihood of a correct match is higher for the residual locations. For individual geoparsing, as only the spatial indicators of the post itself are considered, the scores behave discreet. In contrast, for grouped geoparsing, the scores are averaged between tweets within the same group, and therefore the decrease is more gradual. At very high thresholds, the precision for grouped geoparsing starts to drop (for administrative subdivisions and cities/town/villages). This is likely because the scores assigned to tweets in small groups fluctuate more than for large groups (Sect. 2.2.4) and hence there is more uncertainty in the location being assigned correctly. Therefore, when the

threshold increases, small groups have a larger share in the response set (as large groups will always have averaged medium scores) which causes the precision to drop. Approximately between a threshold of 0.1 and 0.25, precision and recall measures for grouped geoparsing are optimal and higher than using any other threshold for individual geoparsing.

In figure 3, for Toponym recognition, it should be 2.2.1, instead of 2.2.1.

We have updated the section reference accordingly.

Please also note the supplement to this comment:
https://www.nat-hazards-earth-syst-sci-discuss.net/nhess-2017-203/nhess-2017-203-AC1-supplement.pdf

[Figure]

[Figure]

**Fig. 1.** Recall and precision scores for individual and grouped geoparsing with a varying size of the scanning window.

[Figure]

**Fig. 2.** Recall and precision scores for individual and grouped geoparsing with a varying threshold.

---

## Author Comment (AC2) · 30 Aug 2017

We would like to thank the reviewers for his/her very valuable comments. We are happy that the reviewers consider the paper valuable, and we believe the comments have contributed to significant improvements to the manuscript. We have revised the several sections of the paper and added the suggested literature. We also more clearly defined the objectives of our study, and how it differs from existing research. Finally, we followed the instructions of the reviewer, and have improved the (presentation of-) validation and evaluation of TAGGS' performance. Below, we list the reviewer's comments in bold and explain in detail how they have been addressed in the revised paper.

Reviewer: Geotagging and Geoparsing is a well studied problem. The authors have

missed a lot of important work in related work section (see end citations).

We thank the reviewer for pointing out this complimentary literature. We failed to acknowledge the existence of several studies that have addressed geoparsing with a local focus and a priori knowledge of an event. Based on the suggestions of the reviewer, we expanded our literature review substantially. Below, we discuss the most important papers recommended by the reviewer and how they are included in out paper.

Middleton et al., 2014: We thank the reviewer for recommending this paper, which is indeed very relevant. We have included it in our revised paper, particularly focusing on Named Entity Matching. The paper focusses on toponym recognition on a local scale. Using a priori knowledge about the coarse region of the event local data is acquired and used for the toponym recognition. After sub setting the data with the user time zones, NEM is employed, essentially disambiguating toponyms through prioritization based on the scale of the geographical entities. By using this disambiguation technique, NEM improves the precision of geoparsing. We have included the paper in our section discussing related work as follows:

Middleton & Middleton, 2013 show that Named Entity Matching (NEM) performs better than NER on tweets (Middleton et al., 2014). This approach tokenizes tweets and matches these tokens first to places, then streets and finally regions, while discarding matched tokens to avoid double matches.

and

The aim of this study is to develop a global geoparsing algorithm for tweets without assuming a priori knowledge about an event so that the algorithm can be employed for event detection. To geoparse tweets on a global scale without a local focus, additional spatial information from the tweets is required for disambiguation. Therefore, we develop a new toponym disambiguation system which builds upon the approach by Schulz et al., (2013). This new Toponym-based Algorithm for Grouped Geoparsing of Social media (TAGGS) uses grouped geoparsing to reliably find a much larger percentage of locations than does the standard approach of individual geoparsing. The TAGGS approach permits spatial information from related tweets to be incorporated in the analysis, allowing users to geoparse tweets with few or no spatial indicators of their own. TAGGS could form the basis for a global flood detection algorithm in future research. In such study, TAGGS could be combined with other geoparsing algorithms that, after an event is detected and additional geographical data is acquired for a focus area, perform well on a local scale (e.g., Gelernter et al., 2013; Middleton et al., 2013).

- Middleton and Krivcovs, 2016: In this paper, the authors provide a further specification of NEM. Especially the on-demand local geoparsing approach is very interesting and lauded. To employ such an approach, an event detection (automated / manually) step is required, which is the focus of our paper. Once an event is detected such an approach is possible. We revised our text as follows:

The TAGGS algorithm forms the basis for a flood detection algorithm that detects sudden changes in the number of tweets. Once an event is detected, local geoparsing (Gelernter et al., 2013; Middleton et al., 2014) could be employed by on-demand downloading local geographical data to create detailed disaster maps (Middleton et al., 2016).

- Kordopatis-Zilos et al., 2015. We thank the reviewer for bringing this paper to our attention. The paper is particularly focused on geoparsing of photos using a language model and has no particular focus on events. The authors specifically mention that one should be careful to apply such method on volatile events, especially when no extensive training set is available. We have included the paper in our text as follows:

Several approaches exist for the localization of social media posts. Language models typically use a collection of training posts with corresponding geotagged images to determine the most likely location of a new post (Kordopatis-Zilos et al., 2015). However, to apply language models to temporally volatile events, such as floods, a very large training corpus is required. Otherwise, new posts are unlikely to be geotagged to a

location where no event occurred in the training data (Kordopatis-Zilos et al., 2015). Other approaches employ text and/or metadata matching to a gazetteer to detect a user's residence, the location from which the tweet was sent, or the location to which a tweet refers (e.g., Middleton et al., 2013; Schulz et al., 2013).

Reviewer: Although the problem is based on disaster response actually the work presented is standard geoparsing.

As discussed in the introduction of our paper, we focus on the development of a globally applicable event detection methodology for natural disasters in particular. Because of the focus on this specific purpose and to underline the importance relevance of our work to disaster responders, we developed a novel approach for global disaster response using geoparsing. However, we acknowledge that we may have not stated this clearly and therefore added the following to our introduction:

The aim of this study is to develop a global geoparsing algorithm for tweets without assuming a priori knowledge about an event so that the algorithm can be employed for event detection. To geoparse tweets on a global scale without a local focus, additional spatial information from the tweets is required for disambiguation. Therefore, we here develop a new toponym disambiguation system which builds upon the approach by Schulz et al., (2013). This new Toponym-based Algorithm for Grouped Geoparsing of Social media (TAGGS) uses grouped geoparsing to reliably find a much larger percentage of locations than does the standard approach of individual geoparsing. The TAGGS approach permits spatial information from related tweets to be incorporated in the analysis, allowing users to geoparse tweets with few or no spatial indicators of their own.

Reviewer: The location identification approach proposed by the authors is not state of the art (standard tokenization, geonames gazetteer lookup).

Reviewer: The location disambiguation is also quite simplistic (language/time zone/population filters, location name subsumption). Modern geoparsing approaches use extra information to boost precision, either from contextual text (e.g. linguistic patterns used and other location mentions in text), geospatial source (e.g. OpenStreetMap database locations for spatial proximity) or social context (e.g. social media tags from a training corpus). The approach used by the authors is not advancing the state of the art.

Reviewer: The idea of clustering (i.e. grouping) posts with the same location mention to provide additional context is potentially a novel idea that does advance the state of the art.

In this paper we present a new technique for location disambiguation, expanding the approach developed by Schulz et al., (2013) by grouping /clustering tweets to add additional context for use in toponym disambiguation. We thank the reviewer for acknowledging the novelty of our approach.

On the location disambiguation approach, we perhaps failed to highlight the novel aspects. Indeed, as stated above and as discussed in a paper by Middleton & Middleton (2013), several techniques for toponym recognition exist, most importantly, standard tokenization, named entity recognition (NER) and named entity matching (NEM), each including a gazetteer lookup. We develop a new state-of-the-art approach which does not depend on pre-existing knowledge about an event. Because we do not assume a priori knowledge about an event and do not subset the data using filters (e.g., time zone), there is a significant increase in ambiguity of places. Therefore, as recognized in the literature (as we will further discuss below), toponym disambiguation is of great importance compared to geoparsing with a priori knowledge and as such applied with a focus area (Middleton et al., 2016). Hence, our approach heavily focusses on the disambiguation of toponyms after toponym recognition (section 2.2.4 in the paper) by utilizing additional spatial indicators in both the tweet metadata and the text. Therefore, we choose to employ standard tokenization and perform disambiguation using the context of the tweet (text and metadata), rather than prioritization of entities based on their place in the geographical hierarchy (NEM).

We hope that with explanation, it is clearer for the readers where our paper advances the literature. Below, we quote the most important paragraphs from the paper, which summarizes those advances:

We assumed that multiple tweets that mentioned the same toponym within a given timeframe referred to the same location. For example, if a flood occurred in Boston, UK, we expected that all users mentioning "flood" and "Boston" were referencing Boston, UK, rather than Boston, Massachusetts, US. All tweets mentioning the same toponym were then grouped together. Thus, the greater the number of tweets mentioning a location, the larger was the associated group—and therefore, the higher the probability of metadata being available for that group. Since tweets could contain multiple toponyms, individual tweets could belong to more than one group. To continuously geoparse tweets, we used an iterative process (Fig. 4). After the geoparsing of all tweets within the scanning window was finished, the window was shifted by 6 hours. All new tweets were retrieved from the tweet database and separately analyzed for toponyms and respective spatial indicators (Sect. 2.2.1 and 2.2.2), while tweets that fall outside of the scanning window were discarded. The locations mentioned in the tweets within the scanning window were, again, grouped (Sect 2.2.3) and resolved (Sect. 2.2.4) and used to update the toponym resolution table. As the tweet geoparsing process included information from other tweets (including locations referred to in future tweets), it was possible for a tweet's location and respective score to change. In such cases, we updated the database accordingly. Such alterations only occurred when in a subsequent iteration, we found a higher score for a specific location or identified another location with a higher score (but the same toponym). Furthermore, we already use the context (other locations mentioned in the text), as well as other indicators (i.e., user time zones, user home towns and coordinates attached to the tweet) to resolve locations. We do not use these indicators, such as the mentions of related places, as filters but rather as indicators for scoring and subsequent toponym disambiguation. To state this more clearly, we expanded the section on scoring (section 2.2.2) by adding the following text:

For each tweet for which we found one or more candidate locations, as described in Sect. 2.2.1, its additional spatial indicators were matched to each candidate location. We use these indicators as contextual clues to provide additional information for toponym disambiguation.

Reviewer: However the choice to filter out all locations where the population density is small greatly weakens this works applied value. Geoparsing popular locations, usually cities with a high population density, is much easier than geoparsing obscure remote locations. For populous areas social media tagging has been proven to be very effective. Such work (see Kordopatis-Zilos) has not been cited at all and is not compared.

We thank the reviewer for this comment. We agree with the reviewer that areas of low population density are of interest for disaster risk management. However, applying a geoparsing algorithm on a global scale introduces a lot more place names, including many that also have other meanings in normal language usage (e.g., "read", which is both a verb and a small town in the UK). Therefore, we originally included only larger towns. Following the reviewer's suggestions, we have adjusted the algorithm by including small towns, with a population of at least 1, for those tweets in which the town mentioned in the text is written with a capital letter (except for tweets written in German, because in this language all nouns are capitalized). This change increases the number of resolved locations. The revised text now reads:

We consider small towns, with a population of at least 1, if the town mentioned in the text is written with a capital letter. For tweets written in German, we discard all locations with population lower than 5000, because in this language all nouns are capitalized.

We have included Kordopatis-Zilos et al. (2015), as follows:

Several approaches exist for the localization of social media posts. Language models typically use a collection of training posts with corresponding geotagged images to determine the most likely location of a new post (Kordopatis-Zilos et al., 2015). However, this requires access to a very large training corpus when applying language models to

temporally volatile events, such as natural disasters. Otherwise, new posts are unlikely to be geotagged to a location where no event occurred in the training data (Kordopatis-Zilos et al., 2015).

Reviewer: There are publicly available benchmark geoparse datasets (see PyPI geoparsepy https://pypi.python.org/pypi/geoparsepy) which could and should have been used to directly compare results to published work (see Middleton and Gelernter). The authors instead created a smaller dataset manually and thus the results are not directly comparable.

We thank the reviewer for their suggestion of a dataset to use for validation. However, because our approach does not assume a priori knowledge, our work cannot be directly compared against approaches (and datasets) that do assume a priori knowledge. Therefore, we also do not employ techniques to subset the data, for example by selecting data that originates from the time zone of the event. However, since we cannot provide a direct comparison to other work, we have included and updated a comparison between singular and grouped geoparsing to demonstrate that grouped geoparsing indeed performs better than singular geoparsing. We have clarified this distinction and demonstrate the performance increase as follows:

To employ a successful disaster response, it is important that some information about an event is available as soon as possible. The aim of this study is to develop a global geoparsing algorithm for tweets without assuming a priori knowledge about an event. This system can then, in further research, form the basis for a flood detection algorithm and be combined with other geoparsing algorithms that, after additional geographical data is acquired for a focus area, perform well on a local scale. (e.g., Gelernter et al., 2013; Middleton et al., 2013). and Using the manual approach, of the 2,787 total tweets in our validation set, we found 2,035 references to locations in 1,478 tweets. Then, we compared the manually labelled tweets to both the automated individual and automated grouped geoparsing (TAGGS) approaches. For individual geoparsing, we use the location metadata but did not consider other tweets mentioning the same

geographical entities, similar to Schulz et al. (2013).

Reviewer: The evaluation itself is weak. No statistical significance is reported. A breakdown of results by event & language is not provided (known factors in performance). The authors appear to be geoparsing at the region level (the paper is not precise about this so its impossible to tell). The example of 'A10 near Amsterdam' should geoparse both the A10 (road) and Amsterdam (region) - but appears to only score on Amsterdam. The whole approach lacks rigour and depth.

To the best of our knowledge, statistical significance is generally not reported in papers on this topic (e.g., Ju et al., 2016; Middleton et al., 2013).

In our paper, we aim to improve the global disambiguation of toponyms and combine the information of all tweets in all languages. Therefore, we believe that a focus on a specific / location is not a sensible application of our model. We hope we have made this clearer in the revised version of the paper. Hence, the aim of our study is not to improve local geoparsing, which is much better addressed in other studies.

However, when focusing on the performance of the detection of a particular type of event such as floods, a detailed analysis of the performance per country / province / location would be feasible. Unfortunately, the detection of events is beyond the scope of this paper and is discussed as an issue for further study in the recommendations section. However, we do show that there is a trend between the number of geoparsed tweets and the flood damage in a country (Section 3.1). In addition, the performance of a geoparsing algorithm is heavily dependent on a wide range of variables, such as:

- The quality of the underlying gazetteer, which has a great spatial variety. Therefore, when a validation is conducted for an area where the quality of the gazetteer is high (all locations are present, alternative names are available etc.), the validation is likely to yield good results. However, this is in part due to the high quality of the gazetteer, rather than it solely being an indicator of how well-performing our model is (i.e., looking at a particular flood event around Manila would result in a much better performance than by
looking at an event in a remote location in the Philippines.). - The size of the average flood event per country. When a flood event is larger, we expect more tweets related to the disaster. Therefore, more information is available for grouped geoparsing, and thus TAGGS performs better.

Furthermore, as described in the introduction and in Table 3, we geoparse at a country, administrative subdivision and city, town, village-level. To emphasize this, we have moved this statement to the first sentence of the methodology:

The TAGGS algorithm uses geoparsing to match a tweet's location references to one or more geographic locations at a country, administrative subdivision or city, town and village-level.

As stated above, we previously compared the locations of the algorithm to all possible spatial indicators in a tweet, including small-scale features (e.g., restaurants) and used separate references to multiple entities within a higher level geographical entity as one location. We acknowledge that this approach can be confusing. We also note that we excluded several previously included tweets in the manual validation, because for Turkish, we erroneously included tweets mentioning Selçuk of which "sel" (flood in Turkish) is a substring. For these reasons, we re-assessed the validation method, which is now described as follows:

To properly validate the TAGGS algorithm, we defined a golden standard with manually tagged tweets. To the best of our knowledge, no other study provides a global dataset focusing on a specific event type. Therefore, we compile a random dataset using 2,787 flood related tweets from two separate days and manually assign locations to the tweets. Each tweet can be labelled with one, multiple or no locations at all. We recognized all mentions of locations on the different administrative levels that we apply the algorithm to (i.e., country, administrative subdivisions and cities, towns and villages), including abbreviations, shorter versions and slang, but excluded possessive pronouns (e.g., the Irish weather) and mentions of geographical features within towns

and other geographical features, such as valleys and rivers. We do include location mentions when they are combined with other words (e.g., #leedsfloods), but exclude any information in the Twitter handles (e.g., @PakistanToday) because these locations are not necessarily related to the location of a possible event.

Reviewer: The results themselves look weak (i.e precision scores), which is not surprising considering the lack of state of the art location disambiguation techniques. I would expect a precision for region level geoparsing to be 0.9+ (F1 0.8+). The authors manage around 0.7 precision.

We thank the reviewer for their suggestion to use state-of-the-art disambiguation techniques. We assume that the reviewer suggests to employ NEM. As discussed above and clarified in the paper, we do not employ NEM for location disambiguation but rather build upon another technique which uses the text and metadata and consequent grouping of tweets.

Following the reviewer's comments, we have revised major parts of the validation study and updated several components of the model. Originally, we compared the results of our model to the manually labelled validation set on a tweet-by-tweet basis using all types of mentions of locations (e.g., countries, administrative subdivisions, towns, roads, points of interest). We acknowledge that the presentation of the results was somewhat confusing. As is custom in this field, we now present the results of the validation study as recall/precision/F1-measures, split by country, administrative area and town-level. In our paper, we provide a comparison between various model settings (i.e., various thresholds, and a varying size of the scanning window), as shown below:

With geoparsing algorithms, there is a trade-off between the number of tweets that are parsed (recall) and the number of correctly parsed tweets (precision; Leidner, 2007). Precision measures the number of correctly geoparsed tweets relative to the total number of geoparsed tweets. Hence, precision markers do not provide an indication of the total number of tweets within a location. Recall measures reflect the number of correctly geoparsed tweets relative to the total number of tweets with a spatial reference. In essence, the greater the level of precision (i.e., the smaller the number of incorrect tags), the smaller the total number of geoparsed tweets. Inversely, if one wants to geoparse more tweets (higher recall), the number of errors within the geoparsed tweets (in terms of incorrect location assignments) will also increase (lower precision). In the following sensitivity analysis, we show two series of plots (Fig. 1 and 2) delineating both individual (red) and grouped (blue) geoparsing for various model settings namely, a varying threshold and a varying size of the scanning window. In these figures, we show three plots: 1) a plot that shows recall and precision measures for all locations that the model accounts for (i.e., countries, administrative subdivision and cities, towns and villages), using all 2,787 tweets, 2) a plot that shows these measures for administrative subdivisions, using only those tweets that mention such a location according to our validation set, and 3) a plot that shows precision and recall measures for all cities, towns and villages, using only those tweets that mention such a location.

Figure 1 shows the recall and precision scores for individual and grouped geoparsing with a varying threshold. The trade-off between precision and recall is visible in the first window: When a higher threshold is chosen, more location matches are discarded, while the likelihood of a correct match is higher for the residual locations. For individual geoparsing, as only the spatial indicators of the post itself are considered, the scores behave discreet. In contrast, for grouped geoparsing, the scores are averaged between tweets within the same group, and therefore the decrease is more gradual. At very high thresholds, the precision for grouped geoparsing starts to drop (for administrative subdivisions and cities/town/villages). This is likely because the scores assigned to tweets in small groups fluctuate more than for large groups (Sect. 2.2.4) and hence there is more uncertainty in the location being assigned correctly. Therefore, when the threshold increases, small groups have a larger share in the response set (as large groups will always have averaged medium scores) which causes the precision to drop. Approximately between a threshold of 0.1 and 0.25, precision and recall measures for grouped geoparsing are optimal and higher than using any other threshold for individual

geoparsing.

Figure 2 shows the recall and precision measures for a varying scanning window size, ranging between 6 minutes and 48 hours. In theory, when using an infinitesimally small scanning window for grouped geoparsing, the results would be identical to the individual geoparsing. It is clearly visible that, in general, both precision and recall increase when the size of the scanning window is larger. This is expected, because a larger number of tweets are grouped and therefore, the likelihood that spatial information is available increases. Although an increase of recall and precision is still visible for a larger scanning window, the increase is not substantial, which indicates that spatial information is available for most toponyms. When new floods occur, it is not feasible to take location mentions of previous floods into account. Therefore, we hypothesize that when the scanning window becomes too large, the performance of the model will be lower. Unfortunately, because of memory (RAM) constraints in our current setup, we cannot test this. Ideally, the size of the scanning window depends on the volatility of the event type, where events with a longer average duration (people will likely refer to the same event over a longer timespan), such as droughts, could benefit from a larger scanning window and vice versa for shorter events.

While geoparsing without having a priori knowledge about an event and without subsetting the data using filters, introduces much more ambiguity. We derive, in our revised validation study, a precision of approximately 0.9 and a recall of 0.85 (F1 of 0.875) for optimal model settings (threshold 0.2, scanning window 24 hours) for all administrative levels.

Reviewer: Overall this work is too immature at this stage for a journal publication. The lack of a real comparison to the extensive related work already published in this area means this work has little current value to a serious researcher.

As stated above, we acknowledge our paper could be improved, and hence we have revised it using the valuable remarks by the reviewer. We hope that we have addressed

Interactive
comment

the main concerns of the reviewer in this thoroughly revised paper: a clearer presentation of our goals and results, improved references to the existing literature, and a more precise explanation of the novelties in our study and how we build on existing methods. To the best of our knowledge, no other study employed geoparsing for tweets on a global scale for a specific event type, without assuming a priori knowledge about an event. For this reason, we extensively compare our method to the same algorithm, but without the grouping, and show that the grouped geoparsing outperforms singular geoparsing.

The novel grouping approach employed in TAGGS could be combined with other valuable work done on a local scale. For example, using our methods, combined with an event detection algorithm, we can identify regions affected by disasters. As described by Middleton and Krivcovs, 2016, it is then possible to start separate instances of a local approach by acquiring local data (e.g., OpenStreetMap) and employing NEM to further geoparse tweets.

Studies suggested by the reviewer:

Judith Gelernter and Nikolai Mushegian, 2011. Geo-parsing Messages from Microtext. Transactions in GIS, Vol 15, Issue 6, 753–773

Middleton, S.E. Middleton, L. Modafferi, S. "Real-time Crisis Mapping of Natural Disasters using Social Media", Intelligent Systems, IEEE , vol.29, no.2, pp.9,17, Mar.-Apr. 2014

Middleton, S.E. Krivcovs, V. "Geoparsing and Geosemantics for Social Media: SpatioTemporal Grounding of Content Propagating Rumours to support Trust and Veracity Analysis during Breaking News", ACM Transactions on Information Systems (TOIS), 34, 3, Article 16 (April 2016), 26 pages.

Marieke van Erp, Giuseppe Rizzo and Raphaël Troncy, 2013. Learning with the Web: Spotting Named Entities on the intersection of NERD and Machine Learning. In Pro-

ceedings of the 3rd Workshop on Making Sense of Microposts (#MSM2013). Rio de Janeiro, Brazi Alan Ritter, Sam

Clark, Mausam and Oren Etzioni, 2011. Named entity recognition in tweets: An experimental study. In Procedings of Empirical Methods for Natural Language Processing (EMNLP), Edinburgh, UK

Alexandre Davis, Adriano Veloso, Altigran S. da Silva, Wagner Meira Jr., Alberto H. F. Laender, 2012. Named entity disambiguation in streaming data. In Proceedings of the 50th Annual Meeting of the Association for Computational Linguistics, 815 - 824, Jeju, Republic of Korea

Giorgos Kordopatis-Zilos, Symeon Papadopoulos, and Yiannis Kompatsiaris. 2015. Geoparsing social media content with a refined language modelling approach. In Pacific-Asia Workshop on Intelligence and Security Informatics. Ho Chi Minh City, Vietnam. 19 May 2015

References for response:

Gelernter, J., & Balaji, S. (2013). An algorithm for local geoparsing of microtext. GeoInformatica, 17(4), 635–667. https://doi.org/10.1007/s10707-012-0173-8

Ju, Y., Adams, B., Janowicz, K., Hu, Y., Yan, B., & McKenzie, G. (2016). Things and Strings: Improving Place Name Disambiguation from Short Texts by Combining Entity Co-Occurrence with Topic Modeling. In Knowledge Engineering and Knowledge Management: 20th International Conference, EKAW 2016, Bologna, Italy, November 19-23, 2016, Proceedings 20 (pp. 353–367).

Kordopatis-Zilos, G., Papadopoulos, S., & Kompatsiaris, Y. (2015). Geotagging Social Media Content with a Refined Language Modelling Approach. https://doi.org/10.1007/978-3-319-18455-5

Middleton, S. E., & Krivcovs, V. (2016). Geoparsing and Geosemantics for Social Media: Spatiotemporal Grounding of Content Propagating Rumors to Support Trust

and Veracity Analysis During Breaking News. ACM Trans. Inf. Syst., 34(3), 16:1–16:26. https://doi.org/10.1145/2842604

Middleton, S. E., Middleton, L., & Modafferi, S. (2014). Real-time crisis mapping of natural disasters using social media. IEEE Intelligent Systems, 29(2), 9–17. https://doi.org/10.1109/MIS.2013.126

Schulz, A., Hadjakos, A., Paulheim, H., Nachtwey, J., & Mühlhäuser, M. (2013). A Multi-Indicator Approach for Geolocalization of Tweets. Seventh International AAAI Conference on Weblogs and Social Media, 573–582. https://doi.org/papers3://publication/uuid/62449928-74D1-4674-A1A7-24D5F6813F85

Please also note the supplement to this comment:
https://www.nat-hazards-earth-syst-sci-discuss.net/nhess-2017-203/nhess-2017-203-AC2-supplement.pdf

[Figure]

**Fig. 1.** Recall and precision scores for individual and grouped geoparsing with a varying threshold.

[Figure]

**Fig. 2.** Recall and precision scores for individual and grouped geoparsing with a varying size of the scanning window.